# A 5-Years (2015–2019) Control Activity of an EU Laboratory: Contamination of Histamine in Fish Products and Exposure Assessment

**Sonia Lo Magro \*, Simona Summa, Marco Iammarino \*** 📵**, Pasquale D'Antini, Giuliana Marchesani, Antonio Eugenio Chiaravalle** 📵 **and Marilena Muscarella**

Chemistry Department, Istituto Zooprofilattico Sperimentale della Puglia e della Basilicata, via Manfredonia 20, 71121 Foggia, Italy; simona.summa@izspb.it (S.S.); pasquale.dantini@izspb.it (P.D.); giuliana.marchesani@izspb.it (G.M.); eugenio.chiaravalle@izspb.it (A.E.C.); marilena.muscarella@izspb.it (M.M.)
\* Correspondence: sonia.lomagro@izspb.it (S.L.M.); marco.iammarino@izspb.it (M.I.)



**Featured Application: This work offers an overview of the quality of fresh fish and fish products consumed in Puglia and Basilicata regions (south part of Italy), analyzing their histamine content. Indeed, the recent Italian reports released by the Italian Ministry of Health, concerning the official food control activities, indicated the biogenic amines (mainly histamine) as the analyte class with the highest percentage of non-compliant samples, among chemical contaminants. The histamine concentrations detected were elaborated for each type of seafood analyzed, obtaining useful information on the overall quality of fish products. The exposure assessment was also developed. This elaboration gave useful parameters also for other scientists who wish to carry out more extensive risk assessment studies.**

**Abstract:** Histamine contamination was evaluated on 474 batches (3130 determinations) of fish products collected in Puglia and Basilicata (southern part of Italy) during the years 2015–2019, using a high-throughput two-tier approach involving a screening (ELISA test) and confirmatory method (HPLC/FLD with *o-phthalaldehyde* derivatization). Histamine concentration >2.5 mg kg$^{-1}$ was detected in 51% of total batches with the 2.5% of non-compliance. Except for two samples of fresh anchovies, all non-compliant samples were frozen, defrosted and canned tuna. Among 111 fresh tuna batches, 9 had a content of histamine between 393 and 5542 mg kg$^{-1}$, and scombroid poisoning cases were observed after their consumption. Good quality canned tuna and ripened anchovies sold in Italy was observed. Furthermore, the analysis of the processing technology and storage practice critical points were reported in this study, with useful considerations to minimize the histamine risk for consumers. Finally, based on these results, several considerations about risk exposure were reported.

**Keywords:** histamine; scombroid poisoning; official control; enzyme-linked immunosorbent assay; high-performance liquid chromatography; fluorimetric detection; risk exposure

## 1. Introduction

In the last decade, a sharp increase in the marketing and consumption of fresh fish and fish products has been observed [1,2]. EFSA [3] and WHO [4] recommend a consumption of 1–2 fish-based meals per week as fish is essential for a complete diet, due to its high content of proteins, free amino acids and other health-enhancing components such as vitamins, minerals and omega-3 fatty acids [5]. At the same time, the high level of water in the tissues (80%), a low percentage of connective tissue in the muscles and the poorly acidic pH of the tissues make the fish easily subjected to microbial attack and decomposition.

Fish and other seafood may lead to a number of biological and chemical hazards such as pathogenic bacteria, viruses, biotoxins, heavy metals and biogenic amines [6,7]. Histamine, (2-(4-imidazolyl) ethylamine, CAS Number: 51-45-6) (HIM) is a biogenic amine that can be readily produced by bacterial decarboxylases in scombroid and other fish with relatively high free histidine levels in their muscles when alive [8]. HIM production in fish is related to the histidine content of the fish, to the presence of bacterial histidine decarboxylase and to environmental conditions. Scombroid fish belonging to the families *Scombridae* (e.g., tuna and mackerel) and *Scombresosidae* (e.g., saury) but also non-scombroid species belonging to families of *Clupeidae* and *Engraulidae* (e.g., sardines, anchovies, herrings) are characterized by having relatively high levels of histidine in their flesh (from 1 g kg$^{-1}$ in herring to 15 g kg$^{-1}$ in tuna) [9]. The formation of HIM in fish tissue is attributed to a range of microorganisms, such as *Hafnia alvei*, *Morganella morganii*, *Morganella psychrotolerans*, *Photobacterium phosphoreum*, and *Klebsiella pneumoniae* [9,10]. Mesophilic organisms are mostly considered responsible for HIM production at an optimum temperature of 20 to 37 °C, producing more than 500 mg kg$^{-1}$ after 48 h [10]; however, Psychrophilic bacteria (*Photobacterium iliopiscarium*) have been found to produce HIM at levels higher than 500 mg kg$^{-1}$ after culture at 5 °C for 3 days, 10 °C for 36 h, and 20 °C for 18 h [11].

Food quality and freshness has been linked to HIM so that it is possible to use this compound as a fish product quality marker together with other indicators [12]. Ingestion of food containing low amounts of HIM can cause an allergy-like syndrome referred to as "histamine intolerance" [13]. However, in large doses, the normal metabolic mechanisms are insufficient for the detoxification. A HIM intake of 70–1000 mg per single meal may cause so-called "scombroid poisoning" [9], characterized by an incubation period ranging from a few minutes to hours and symptoms such as headache, nasal secretion, hypotension, tachycardia and possible death in sensitive subjects [14]. Safety requirements about HIM presence in seafood and rules for correct sampling were fixed by the European Commission in the Regulation (EC) No. 2073/2005 and subsequent modifications [15]. Sampling must be in nine units per batch, taking into account the inhomogeneous distribution of the toxin in the fish [16]. Therefore, nine independent analyses are needed to assess the compliance of a single batch product. For fishery products obtained from fish species with a high amount of histidine, (particularly the families *Scombridae, Clupeidae, Engraulidae, Coryfenidae, Pomatomidae,* and *Scombresosidae*) among the nine sample units analyzed for each batch, the mean value must not exceed 100 mg kg$^{-1}$ (m), two may have a value of more than 100 mg kg$^{-1}$ (m) but less than 200 mg kg$^{-1}$ (M) and no sample unit may have a value exceeding 200 mg kg$^{-1}$ (M). Higher HIM levels, not more than twice the above values, are permitted in fishery products which have undergone enzyme ripening treatment in brine. In addition, a further modification of Regulation (EC) No. 2073/2005 [17] sets less strict rules for sampling at the retail level: a single sample unit may be taken and analyzed and the whole batch is judged unsafe only if the result is found to be above M. In the last five years, the RASFF notifications [18] regarding the presence of HIM above the permitted limits concern not only a "historical" matrix such as those ripened in brine anchovies, but also (and more frequently) fresh, frozen and defrosted tuna of EU and extra-EU origin.

The purchase of defrosted fish has increased considerably in recent years due to its lower price compared to fresh fish. In this matter, the sale of defrosted products labeled as fresh is difficult to "unmask" and is not uncommon [19]. The defrosted fish should be consumed within 24 h from the first defrosting and must not be re-frozen. Non-observance of these indications can be a serious health consumer risk, especially in the case of tunas already containing a certain amount of HIM which may increase over time or during the frosting/defrosting phases. Another critical point for the HIM formation is the storage method on the retail market. Tuna as strands or slices, anchovies and the others kinds of fresh fish are commercialized not refrigerated but are displayed lying on ice. In this condition the fish, being only partially in contact with ice, are exposed at room temperature, accelerating the bacterial spoilage and HIM formation. Furthermore, in 2016, the Health and Food Safety Commission was informed about the supposed treatment of tuna destined for canning with vegetable extracts containing a high concentration of nitrites/nitrates [20]. These additives, in breach of the specifications laid in Commission Regulation (EC) n. 231/2012 [21], conferred a red color to the fish

tissues, masking deterioration and HIM formation. In the spring of 2017, more than 150 people in Spain were affected by scombroid poisoning after tuna consumption [20]. In the Puglia and Basilicata regions (south part of Italy), in the same period, several cases of intoxication after the consumption of tuna-based meals were reported. Tuna samples used for meal preparation, seized by competent authorities and analyzed in our laboratory were found to be non-compliant to the HIM limits fixed by EC. These cases are discussed in the present study, which is an overview of the quality of fresh fish and fish products consumed in the Puglia and Basilicata regions. In this study, a contribution to the evaluation of exposure to HIM from fish consumption is reported. The results obtained by analyzing fresh, canned and ripened fish products for the determination of HIM are described and elaborated as risk exposure. The analyses were carried out within the official control plans in charge to the Istituto Zooprofilattico Sperimentale della Puglia e della Basilicata (IZS-PB) in the last five years (2015–2019), by using validated and accredited analytical techniques.

## 2. Materials and Methods

### 2.1. Sample Collection

Data about the content of HIM in fish products were obtained from official control analyses performed on 474 batches (3130 determinations) of imported and national fish products collected in Puglia and Basilicata (Italy) in the years 2015–2019. Generally, each batch was collected by the technicians of the local Health Service and border control authorities in nine sample units, as indicated in the Commission Regulation (EC) 2073/2005 [15]. However, in specific cases (suspected poisoning episode or sampling in the retail market) only a sample unit was taken. The samples were subdivided into three categories: 294 (62%) fresh/frozen and defrosted products—yellowfin tuna (*Thunnus albacares*, Bonnaterre 1788), skipjack tuna (*Katsuwonus pelamis*, Linnaeus 1758), bluefin tuna (*Thunnus thynnus*, Linnaeus, 1758), anchovies (*Engraulis encrasicolus*, Linnaeus, 1758), mackerel (*Scomber scombrus*, Linnaeus, 1758), sardines (*Sardina pilchardus*, Walbaum 1792), herring (*Clupea harengus*, Linnaeus, 1758), other kind of fish as swordfish (*Xiphias gladius* Linnaeus, 1758), salmon (*Salmo salar*, Linnaeus, 1758), and other species as gadidae and cephalopods; 119 (25%) canned products (mainly tuna and mackerel), and 61 (13%) ripened products (mainly anchovies and sardines). Canned tuna samples—made with skipjack tuna or yellowfin tuna—and canned mackerel samples, in water or in oil, were stored at room temperature until analysis. The majority of tuna samples were in 80 g cans while mackerel samples were in 125 g cans. The only non-compliant sample among the canned products was commercialized in a 2500 g can. The fresh fish samples were brought to the laboratory in cold storage conditions and stored at −20 °C until analysis.

The dietary exposure estimates are usually derived from the minimum, mean and maximum residue levels found in each category of product monitored. These levels, combined with representative data about food consumption, generate three different exposure scenarios (low, average and high) that represent the likely exposure across a population [22].

During the survey, different samples with high HIM concentrations (up to 5542 mg kg$^{-1}$) were registered. Obviously, these high concentrations are responsible for acute poisoning and the risk assessment under high exposure scenario loses its sense. Consequently, the risk assessment was developed taking into account the average exposure scenario. Moreover, in order to give a more significant contribution to the evaluation, the HIM concentrations detected during monitoring were subdivided into 6 ranges (C ≤ 2.5 mg kg$^{-1}$, 2.5 mg kg$^{-1}$ < C ≤ 10 mg kg$^{-1}$, 10 mg kg$^{-1}$ < C ≤ 50 mg kg$^{-1}$, 50 mg kg$^{-1}$ < C ≤ 100 mg kg$^{-1}$, 100 mg kg$^{-1}$ < C ≤ 200 mg kg$^{-1}$, C > 200 mg kg$^{-1}$) and the risk assessment was also developed taking into account the mean concentration registered in the range with the highest amount of data (most likely scenario).

All samples monitored during this survey were commercialized in Italy. So, the reference data about fish consumption were obtained from the INRAN-SCAI 2005-06 [23–25], which provides the mean consumption of the Italian population relating to the following categories of seafood: fresh and

frozen, preserved (canned and ripened fish) and overall. For each category, data on 5 subgroups of the population were available: infants (0–2 a), children (3–9 a), adolescents (10–17 a), adults (18–64 a), the elderly (65–97 a). Among these 5 subgroups, only the last 4 were taken into account for risk assessment, since the estimations relating to the first one (infants) was not representative (*n* < 30) [26]. The risk assessment was elaborated using data related to the whole population and to the seafood consumers, since the reference document supplied this type of information.

A special focus was reserved for canned tuna, which is the fish species mainly analyzed in this survey and, at the same time, the most consumed marine species in Europe, with an average annual consumption equal to 2.78 kg per capita in 2016, corresponding to 11% of all seafood consumed [27]. Given the Italian consumption of canned tuna, recently estimated at 2.5 kg per capita [28], the exposure assessment was also carried out for canned tuna consumption.

For estimating the exposure and the resultant risk, under a probabilistic approach, the HIM no- observed-adverse-effect level (NOAEL) was taken as reference. As reported by several official documents, studies and scientific opinions, for HIM this level corresponds to 50 mg $Die^{-1}$ [29–31].

### 2.2. Sample Preparation

The sample preparation was related to the type of sample analyzed. For fresh fish, the skin and bones were removed just before homogenization. Frozen fish was left to defrost at 4 °C and then prepared. Canned fish samples were dried on the absorbent paper to remove as much preserving liquid as possible (oil or water). For salted herrings and anchovies, the salt present in large grains in the sample was removed with the aid of a sharp knife and of a paper cloth. Once the salt was removed, the bones were taken off. To avoid the increase in HIM levels, the fish was kept at room temperature in the strictly necessary time for the sample pre-treatment. The knives used for fish pre-treatment were cleaned with ethanol to avoid contamination of the samples.

### 2.3. ELISA Test Reagents, Equipment and Procedure

Screening analyses were performed using a Neogen Veratox® Histamine kit. All reagents used for the ELISA screening analysis were included in the commercial kit. Sample preparation was carried out according to the kit instructions with minor changes. Briefly, the pre-treated sample was homogenized using a commercial blender. One gram of each aliquot was transferred into a 15 mL centrifuge polypropylene tube and 9 mL of distillate water was added. After shaking at room temperature for 10 min, the mixture was centrifuged at 2112 g and 10 °C for 10 min; 100 μL of the aqueous supernatant was diluted with 10 mL of kit diluent buffer. A further 5-fold dilution was necessary to make the concentration compliant to the standards curve. Thus, the sample was ready to be analyzed according to the instructions reported in the manufacturer's kit manual.

A microtiter plate spectrophotometer (Anthos HT2, GSG ROBOTIX s.r.l. (Cinisello Balsamo, Milano, Italy) was used for reading the ELISA plates at 620 nm. A calibration curve was obtained, plotting absorbance values against the concentration of standard solutions at 0, 20, 50 mg $kg^{-1}$. Sample concentration values were multiplied to the dilution factor of five. In each analytical batch, positive and negative quality control samples were included. The negative quality control was a blank fish (i.e., tuna and/or ripened anchovies) depending on the type of samples analyzed in the daily batch, while the positive quality control was the same blank tuna spiked at 100 mg $kg^{-1}$ and/or the same blank ripened anchovies spiked at 200 mg $kg^{-1}$. Unknown samples, standard solutions and quality control were pipetted twice into the microplate wells. Elisa test, allowing the detection of HIM content at concentrations between 2.5 and 250 mg $kg^{-1}$, was fully validated. Details of validation are reported elsewhere [32].

Samples analyzed by the ELISA method with an HIM content greater than 77 mg $kg^{-1}$ (Maximum Limit—2 × relative standard deviation of the screening method repeatability) for fresh and canned fish and 177 mg $kg^{-1}$ (Maximum Limit—2 × relative standard deviation of the screening

method repeatability) for processed fish were judged "suspicious non-compliant" samples and they were subsequently analyzed by an HPLC/FLD method with online derivatization with *o-phthalaldehyde*.

## 2.4. HPLC Reagents, Equipment, and Procedure

The standard of histamine dihydrochloride (≥99%), sodium 1-decanesulfonate (≥99.0%), potassium phosphate bibasic (≥98.0%), and potassium phosphate monobasic (≥98.0%) was supplied by Sigma–Aldrich (Steinhem, Germany). Trichloroacetic acid (TCA) of ACS grade (99%) was purchased from Carlo Erba Reagents (Rodano, Italy). Water and acetonitrile of HPLC grade were purchased from Baker (Deventer, The Netherlands). The derivatization reagents *o-phthalaldehyde*, N,N-dimethyl-2-mercaptoethylamine (Thiofluor™) and potassium borate buffer (*o-phthalaldehyde* diluent, OD104) were from Pickering Laboratories (Mountain View, CA, USA).

A 1000 mg L$^{-1}$ stock solution of HIM was prepared in water and stored at +4 °C for up to 4 months. A 10 mg L$^{-1}$ solution was freshly prepared in 5% TCA. Working standard solutions (0.04, 0.5, 1, 4, 8 mg L$^{-1}$) were obtained by dilution with 5% TCA, just before analysis.

Next, 1.70 g of potassium phosphate monobasic, 2.04 g of potassium phosphate bibasic, and 0.49 g of sodium 1-decanesulfonate were dissolved in water for the preparation of the phosphate buffer solution, containing an ion-pair reagent. The pH value of this solution was adjusted to 5.40 ± 0.05 with concentrated hydrochloric acid and the volume was made up in a 1 L flask with water. The post-column derivatization solution was freshly prepared by dissolving 0.2 g of Thiofluor™ in about 1 mL of OD104 and was added to a solution consisting of 0.2 g of *o-phthalaldehyde* in about 1 mL of methanol; the mixture was adjusted to a final volume of 200 mL by OD104. This solution was stored in a brown glass bottle and, when not in use, was kept at 4 °C for up to 2 days.

Chromatographic separations were performed on an HPLC system, Agilent Technologies SL 1200 Series (Waldbronn, Germany) consisting of a binary pump provided with a micro vacuum degasser, a thermostatable autosampler, a column compartment, and a fluorescence detector.

Online post-column chemical derivatization was performed by using a commercially available system supplied by LCTech GmbH (Dorfen, Germany) and consisting of a double-piston pump (model K-120) and a thermostatable post-column reactor (model CRX 400) equipped with a 0.5 mL knitted reaction coil. Separations were performed using Phenomenex columns: Luna C8 column (250 mm × 4.6 mm i.d., particle size 5 µm) at a flow rate of 1.0 mL min$^{-1}$ by isocratic elution. The mobile phase consisted of 0.01 M phosphate buffer solution, containing 0.002 M sodium 1-decanesulfonate as ion-pair reagent, at pH 5.4 (A) and acetonitrile (B) in a ratio of 80:20 (*v/v*). The injection volume was 2 µL and the column temperature was set at 40 °C. The output of the column was connected by a T-joint with the derivatization line, pumping at a flow rate of 0.4 mL min$^{-1}$, and carried to the reaction coil, set at the temperature of 40 °C, for the derivatization process. Fluorescence detection was performed at the excitation and emission wavelengths of 343 and 445 nm, respectively. The system was interfaced, via network chromatographic software (Agilent ChemStation), to a personal computer to control instruments, data acquisition and processing.

A 5 g portion of pre-treated sample was added to 30 mL of 5% TCA and the mixture was homogenized, in a single-step, with the aid of IKA ULTRA-TURRAX T18 basic (Werke GMBH & Co. KG, Germany). After centrifugation at 2112 g for 10 min at 10 °C and separation of the supernatant, few milliliters of 5% TCA was added to the sample extract, to obtain a final volume of 30 mL. An aliquot of 1 mL of this solution was filtered with an Anotop 10 LC filter (0.2 µm, 10 mm, Whatman) obtaining an extract stable for 3 days at +4 °C. An amount of 100 µL of this extract was diluted with 900 µL of water and then injected in HPLC. The procedure resulted in a 60-fold dilution of HIM, evaluated against a known concentration added to a real blank sample. Likewise, in the screening test, in each confirmatory batch of the method, both positive and negative quality control samples were included. Together with the screening suspect samples, a blank and a spiked sample fortified at regulatory limit (i.e., 100 or 200 mg kg$^{-1}$ depending on the type of fish product analyzed) were run. Each sample unit was analyzed in duplicate and the results were calculated as an average of respective replicates. In 2017,

the instrumental method (HPLC/FLD) used in this study was compared to the EN ISO 19343: 2017 [33] obtaining better or comparable analytical performances (i.e., limit of quantification and precision).

### 2.5. Methods Quality Control

ELISA and HPLC/FLD methods, fully validated and accredited by the Italian Organism for laboratories accreditation ACCREDIA, are currently used in our laboratory in routine analyses and their reliability is checked every year by participating in proficiency tests. This external quality control involves the analysis of two unknown lyophilized tuna muscle samples "incurred" with HIM at two different levels. The Z-scores obtained in 2019 were equal to 1.10–0.53, and 1.11–1.22 for the ELISA and HPLC method, respectively, confirming the suitability of both procedures for official control.

### 2.6. Data Handling and Statistical Analysis

The samples resulted as contaminated if the HIM concentration was above 2.5 mg kg$^{-1}$, which is the limit of quantification of both ELISA and HPLC/FLD methods, used respectively for screening and confirmatory analysis. In the case of sampling and analysis of nine unit samples (332 cases on 474 batches), the HIM content was evaluated considering the mean value, replacing with LOQ/2 values below LOQ. This approach was used since it represents a balance point between the solution that underestimates (=0) or overestimates (=LOQ) the true value, and it is reasonably precautionary in health-related studies [34]. One-way ANOVA and Student's *t*-test ($p < 0.05$) were used to compare the contamination levels of the different types of seafood investigated. Statistical handling of the results was performed using the Excel Microsoft Office software version 2016 for Windows.

## 3. Results

The overall results are reported in Figure 1, which shows the number of total samples associated with the different concentration ranges. These ranges were set taking into account both the compliance limits m and M (as reported in the Regulation (EC) No. 2073/2005) and the different levels of fish freshness, as follows: C < 10 mg kg$^{-1}$ = fish of good quality; 10 < C < 50 mg kg$^{-1}$ = significant deterioration; C > 50 mg kg$^{-1}$ = evidence of definitive decomposition [35,36]. HIM level > 2.5 mg kg$^{-1}$ was detected in 242 of the 474 analyzed samples (51%), with 12 non-compliant responses (2.5%). HIM concentration less than 10 mg kg$^{-1}$ was detected in 410 samples (87% of the total).

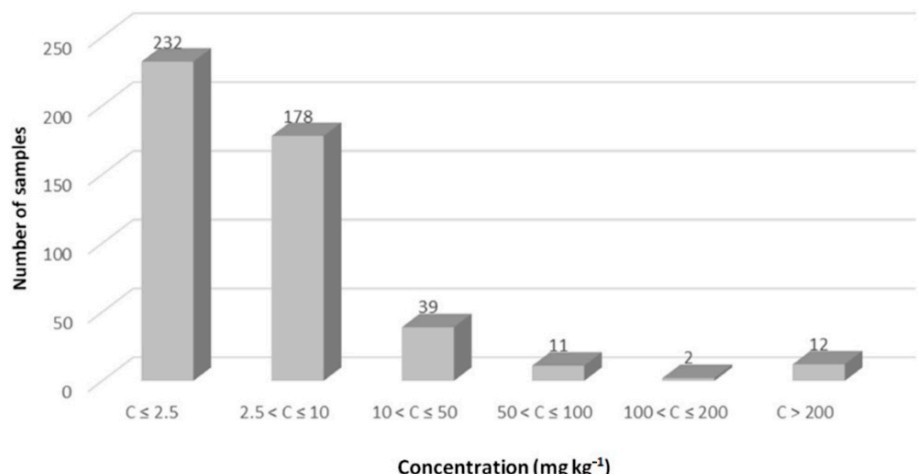

**Figure 1.** Distribution of HIM content in overall fish products. C = HIM concentration (mg kg$^{-1}$).

Figure 2 shows HIM concentration ranges in fish categories (fresh/frozen and defrosted products, canned products, ripened products). For fresh/frozen and defrosted category, 260 samples out of 294 analyzed showed an HIM concentration below 10 mg kg$^{-1}$ and 11 non-compliant samples out of the 294 analyzed were detected. On a total of 119 batches of canned samples, 117 contained HIM levels

below 50 mg kg$^{-1}$, with 110 samples below 10 mg kg$^{-1}$. Finally, as concerns ripened products samples (61 on a total of 474), HIM levels less than 50 mg kg$^{-1}$ were detected in 55 samples.

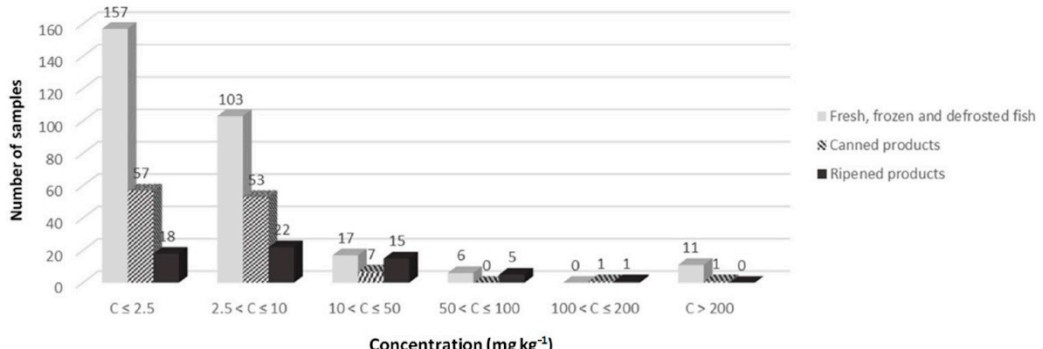

**Figure 2.** Distribution of HIM content in fish categories (fresh/frozen/defrosted fish, canned products, ripened products). C = HIM concentration (mg kg$^{-1}$).

In Table 1 is reported the HIM content in mg kg$^{-1}$ (mean, minimum and maximum) for each kind of product. From a statistical analysis, a significant difference was verified among different seafood types, using one-way ANOVA ($p < 0.05$). The highest HIM levels were quantified in fresh/frozen/defrosted tuna. The results of these levels were statistically higher than in those detected in canned tuna and other fresh fish and were comparable to the concentrations quantified in ripened anchovies, fresh/frozen mackerel, canned mackerel, and fresh/frozen sardines and herrings (*t*-test, $p < 0.05$). Table 1 also reports the number and the concentration levels of non-compliant batches, ranging from 328 to 5542 mg kg$^{-1}$. Finally, the exposure assessment from seafood consumption, under an average exposure scenario (Table 2) and under the most likely exposure scenario (Table 3), are reported.

**Table 1.** Histamine content in mg kg$^{-1}$ (mean, minimum, and maximum) for each product type and number of non-compliant batches.

| Fish Species | Products | Number of Batches | Histamine Content (mg kg$^{-1}$) | | | Non-Compliant Batches | Histamine Content in Non-Compliant Batches (mg kg$^{-1}$) |
|---|---|---|---|---|---|---|---|
| | | | Mean | Min * | Max | | |
| Tuna | Fresh/frozen/defrosted | 111 | 197 [a] | 2.6 | 5542 | 9 | 393–4895–695–2250–443–887–5542–1628–4532 |
| | Canned | 100 ** | 4.9 [b,c] | 2.6 | 38 | 0 | - |
| Anchovies | Fresh/frozen | 62 | 20 [b] | 2.6 | 559 | 2 | 328–559 |
| | Ripened | 57 | 14 [a,b] | 2.7 | 132 | 0 | - |
| Mackerel | Fresh/frozen | 45 | 7.8 [a,b] | 2.6 | 68 | 0 | - |
| | Canned | 12 | 11 [a,b] | 3.0 | 105 *** | 0 | - |
| Sardines and herrings | Fresh/frozen | 14 | 2.7 [a,b] | 2.8 | 5.6 | 0 | - |
| | Ripened | 4 | 19 **** | 3.7 | 55 | 0 | - |
| Other fresh fish | - | 62 | 3.1 [b,c] | 2.6 | 3.2 | 0 | - |
| Other canned products | - | 6 | 2.7 **** | 4.6 | 6.4 | 0 | - |

* Minimum value of HIM concentration found above the LOQ; ** A non-compliant sample residual of the meal (HIM content of 2219 mg kg$^{-1}$) deriving from an open can was not included; *** Measurement uncertainty: 13 mg kg$^{-1}$; **** *t*-test not significant ($n \leq 6$); Mean values with different superscript significantly differ ($p < 0.05$).

**Table 2.** Histamine risk exposure from seafood consumption—Average exposure scenario.

| | Fresh and Frozen Seafood * | | | | | |
|---|---|---|---|---|---|---|
| **Population Group** | **Global Population mg Die$^{-1}$ (% of NOAEL)** | | | **Only Consumers mg Die$^{-1}$ (% of NOAEL)** | | |
| | **Male** | **Female** | **Male + Female** | **Male** | **Female** | **Male + Female** |
| Children (3–9 a) | 3.7 (7.4%) | 2.5 (5.0%) | 3.2 (6.4%) | 5.3 (10.6%) | 4.2 (8.4%) | 4.7 (9.5%) |
| Adolescents (10–17 a) | 3.9 (7.7%) | 3.5 (7.1%) | 3.9 (7.9%) | 6.0 (12.1%) | 5.5 (10.9%) | 5.7 (11.4%) |
| Adults (18–64 a) | 3.9 (7.7%) | 3.1 (6.3%) | 3.7 (7.4%) | 5.6 (11.1%) | 5.2 (10.5%) | 5.4 (10.8%) |
| Elderly (65–97 a) | 3.9 (7.7%) | 2.3 (4.7%) | 3.1 (6.3%) | 5.4 (10.8%) | 4.5 (9.0%) | 4.9 (9.8%) |

| | Preserved Seafood * | | | | | |
|---|---|---|---|---|---|---|
| **Population Group** | **Global Population mg Die$^{-1}$ (% of NOAEL)** | | | **Only Consumers mg Die$^{-1}$ (% of NOAEL)** | | |
| | **Male** | **Female** | **Male + Female** | **Male** | **Female** | **Male + Female** |
| Children (3–9 a) | 0.03 (0.05%) | 0.03 (0.07%) | 0.03 (0.05%) | 0.09 (0.19%) | 0.09 (0.19%) | 0.09 (0.19%) |
| Adolescents (10–17 a) | 0.03 (0.07%) | 0.04 (0.09%) | 0.04 (0.09%) | 0.15 (0.29%) | 0.13 (0.26%) | 0.14 (0.28%) |
| Adults (18–64 a) | 0.06 (0.12%) | 0.05 (0.10%) | 0.06 (0.12%) | 0.16 (0.33%) | 0.15 (0.31%) | 0.16 (0.33%) |
| Elderly (65–97 a) | 0.05 (0.10%) | 0.03 (0.07%) | 0.04 (0.09%) | 0.16 (0.33%) | 0.15 (0.29%) | 0.15 (0.31%) |

| | Seafood—Overall * | | | | | |
|---|---|---|---|---|---|---|
| **Population Group** | **Global Population mg Die$^{-1}$ (% of NOAEL)** | | | **Only Consumers mg Die$^{-1}$ (% of NOAEL)** | | |
| | **Male** | **Female** | **Male + Female** | **Male** | **Female** | **Male + Female** |
| Children (3–9 a) | 2.7 (5.3%) | 2.0 (4.0%) | 2.3 (4.6%) | 3.8 (7.6%) | 3.0 (6.0%) | 3.4 (6.8%) |
| Adolescents (10–17 a) | 2.8 (5.5%) | 2.8 (5.7%) | 2.8 (5.7%) | 4.3 (8.7%) | 3.9 (7.9%) | 4.1 (8.2%) |
| Adults (18–64 a) | 2.8 (5.5%) | 2.6 (5.2%) | 2.7 (5.3%) | 4.0 (8.0%) | 3.8 (7.5%) | 3.9 (7.7%) |
| Elderly (65–97 a) | 2.8 (5.5%) | 2.9 (3.8%) | 2.3 (4.5%) | 3.9 (7.7%) | 3.2 (6.5%) | 3.5 (7.1%) |

* Source: INRAN-SCAI 2005-06. https://www.crea.gov.it/web/alimenti-e-nutrizione/-/indagine-sui-consumi-alimentari.

**Table 3.** Histamine risk exposure from seafood consumption—Most likely exposure scenario.

| | **Fresh and Frozen Seafood *** | | | | | |
| --- | --- | --- | --- | --- | --- | --- |
| **Population Group** | **Global Population mg Die$^{-1}$ (% of NOAEL)** | | | **Only Consumers mg Die$^{-1}$ (% of NOAEL)** | | |
| | **Male** | **Female** | **Male + Female** | **Male** | **Female** | **Male + Female** |
| Children (3–9 a) | 0.24 (0.48%) | 0.18 (0.36%) | 0.21 (0.42%) | 0.34 (0.69%) | 0.27 (0.54%) | 0.31 (0.61%) |
| Adolescents (10–17 a) | 0.25 (0.50%) | 0.25 (0.51%) | 0.25 (0.51%) | 0.39 (0.78%) | 0.35 (0.71%) | 0.37 (0.74%) |
| Adults (18–64 a) | 0.25 (0.50%) | 0.23 (0.47%) | 0.24 (0.48%) | 0.36 (0.72%) | 0.34 (0.68%) | 0.35 (0.70%) |
| Elderly (65–97 a) | 0.25 (0.50%) | 0.17 (0.34%) | 0.20 (0.41%) | 0.35 (0.70%) | 0.29 (0.58%) | 0.32 (0.63%) |
| | **Preserved Seafood *** | | | | | |
| **Population Group** | **Global Population mg Die$^{-1}$ (% of NOAEL)** | | | **Only Consumers mg Die$^{-1}$ (% of NOAEL)** | | |
| | **Male** | **Female** | **Male + Female** | **Male** | **Female** | **Male + Female** |
| Children (3–9 a) | 0.02 (0.03%) | 0.02 (0.04%) | 0.02 (0.03%) | 0.06 (0.12%) | 0.06 (0.12%) | 0.06 (0.12%) |
| Adolescents (10–17 a) | 0.02 (0.04%) | 0.03 (0.06%) | 0.03 (0.06%) | 0.09 (0.19%) | 0.08 (0.17%) | 0.09 (0.18%) |
| Adults (18–64 a) | 0.04 (0.08%) | 0.03 (0.07%) | 0.04 (0.08%) | 0.10 (0.21%) | 0.10 (0.20%) | 0.10 (0.21%) |
| Elderly (65–97 a) | 0.03 (0.07%) | 0.02 (0.04%) | 0.03 (0.06%) | 0.10 (0.21%) | 0.09 (0.19%) | 0.10 (0.20%) |
| | **Seafood—Overall *** | | | | | |
| **Population Group** | **Global Population mg Die$^{-1}$ (% of NOAEL)** | | | **Only Consumers mg Die$^{-1}$ (% of NOAEL)** | | |
| | **Male** | **Female** | **Male + Female** | **Male** | **Female** | **Male + Female** |
| Children (3–9 a) | 0.25 (0.49%) | 0.19 (0.38%) | 0.21 (0.43%) | 0.35 (0.71%) | 0.28 (0.56%) | 0.32 (0.63%) |
| Adolescents (10–17 a) | 0.26 (0.51%) | 0.26 (0.53%) | 0.26 (0.53%) | 0.40 (0.80%) | 0.36 (0.73%) | 0.38 (0.76%) |
| Adults (18–64 a) | 0.26 (0.51%) | 0.24 (0.48%) | 0.25 (0.49%) | 0.37 (0.74%) | 0.35 (0.70%) | 0.36 (0.72%) |
| Elderly (65–97 a) | 0.26 (0.51%) | 0.18 (0.35%) | 0.21 (0.42%) | 0.36 (0.72%) | 0.30 (0.60%) | 0.33 (0.65%) |

* Source: INRAN-SCAI 2005-06. https://www.crea.gov.it/web/alimenti-e-nutrizione/-/indagine-sui-consumi-alimentari.

## 4. Discussion

### 4.1. HIM Levels in Fish Samples

The screening/confirmation double approach adopted in this work was successfully applied, especially in preventive seizure cases, to check quickly and reliably the conformity of fish batches. The high methods sensitivity (quantification limit: 2.5 mg kg$^{-1}$) and the good precision (RSDr < 10% for ELISA test and RSDR < 5% for HPLC/FLD method) allowed us to quantify HIM over the whole contamination range, generating useful data for the exposure assessment. The results showed a

lower occurrence of non-compliant samples (2.5%) in comparison to data found in the literature in older monitoring, where a 5% and 4.9% non-compliance percentage was found [32,37]. Instead, Cicero et al. [38] reported in a recent study a comparable rate of non-compliance, of about 2.79%. Finally, HIM concentration less than 10 mg kg$^{-1}$, detected in the 87% of the total samples, indicated an overall good quality of the analyzed products in comparison than that reported by Petrovic et al. [39] about fish products imported into Serbia (67.03% on a total of 273 samples below 10 mg kg$^{-1}$).

### 4.1.1. Fresh, Frozen, and Defrosted Fish

As shown in Figure 2, the fresh, frozen, and defrosted fish category, mainly represented by tuna, anchovies, and mackerel, showed a particular situation: the highest number of samples with concentrations below 10 mg kg$^{-1}$. At the same time, the highest number of "non-compliant" samples was found in this category. The non-compliances concerned nine frozen and defrosted tuna samples and two anchovie samples with concentration values from 328 to 5542 mg kg$^{-1}$, much higher than the limits set in the Regulation (EC) No. 2073/2005 (Table 1). Critical points such as inadequate frosting/defrosting cycles, incorrect treatment before consumption of the defrosted tuna, and inappropriate packaging or storage conditions at the retail market are, presumably, at the basis of these findings in agreement with those reported by Altieri et al. and Đorđević et al. [40,41] Furthermore, fraudulent practices in the treatment of raw tuna with red extracts containing nitrates and/or high levels of antioxidants or with carbon monoxide, as reported in the last years [20], may be other causes of the HIM presence. As shown in Table 1, in the fresh, frozen and defrosted fish category, tuna showed the highest HIM concentrations, in accordance with most of the studies reported in the literature [38,42–44], confirming that tuna is more susceptible to HIM development than mackerel, anchovies, and sardines because of its high content of free histidine and its composition and presence of high levels of bacterial flora. As reported in Table 1 for fresh anchovy samples, an HIM content in the range 2.57–559 mg kg$^{-1}$ was found, with a mean concentration of 19.6 mg kg$^{-1}$. This value was lower than that of 41.1 mg kg$^{-1}$ reported by Park et al. [45] In the same study, the authors found a mean HIM concentration of 39.3 mg kg$^{-1}$ on 30 frozen mackerel lots. This value was higher than that of 7.8 mg kg$^{-1}$ detected, in the present survey, in fresh/frozen mackerel samples that showed also HIM levels ranging to 2.57 from 68.4 mg kg$^{-1}$. The HIM levels found in mackerel were also much lower than those reported by Ali et al. in Indian mackerels (*Rastrelliger kanagurta*) of Pakistan (144.72 ± 2.47 mg kg$^{-1}$) [46]. Finally, our results are quite similar to those reported by Cicero et al. for Mediterranean mackerel sampled in the years 2010–2012 but were much lower with respect to the HIM values found in the same kind of product analyzed in 2014 [38].

HIM concentrations less than 10 mg kg$^{-1}$ were detected in non-scombroid fish such as salmon and other species for which no limits were set in the Regulation (EC) No. 2073/2005. These findings are consistent with the literature data [47] even though salmon consumption is reported as a cause of intoxication [48,49] and recent studies showed the presence of HIM in non-scombroid species such as mahi-mahi and swordfish in fillets [50,51].

Two samples of tuna slices stored in the freezer, taken from a local restaurant and from a company canteen, collected after allergic reaction cases, showed HIM concentration levels of 4895 mg kg$^{-1}$ and 5542 mg kg$^{-1}$, respectively. Restaurants, company canteens and sandwich shops are often reported as sources of scombroid poisoning cases [52]. Apart from frozen or defrosted tuna, used in restaurants for meal preparation but previously stored in inappropriate conditions and not consumed within 24 h, canned tuna is also used as an ingredient in salad, sandwiches and pizza, and when opened a long time before consumption, can be subject to bacterial contamination and HIM production as reported by Cattaneo et al. [53].

### 4.1.2. Canned Products

As previously reported in the Results section (Figure 2), on a total of 119 batches of canned samples, 98% of the samples contained HIM levels below 50 mg kg$^{-1}$, with 92% of the samples below 10 mg kg$^{-1}$.

A number of 91 in a total of 101 canned tuna samples were processed by Spanish companies destined for fish transformation with raw tuna of EU and extra-EU origin. The majority of canned mackerel (9 samples out of 12 analyzed) were produced in Morocco. In this country, some critical issues in the canned product processing chain were verified over the past few years. Tunas and mackerels are often caught far from canning factories and so an inadequate interim conservation can lead to bacterial spoilage and HIM formation. Once formed, there is no method of fish preparation available, including thermic treatment, which can degrade the toxin. Therefore, a low content of HIM in the canned analyzed samples could indicate a general improvement in the production chain of canned products, coming from good quality of raw material, a correct maintenance of fish cold chain and adequate handling in the working place and in the processing [54]. No appreciable differences in HIM contamination ($p < 0.05$) were noted between samples of tuna in different preserving liquid (in oil or natural) as already reported by Sadeghi et al. [55] Good quality canned tuna sold in Italy has already been reported in the literature [32,37,38]. Our results are comparable with those reported by Silva et al. [56] with an HIM mean level for canned tuna products marketed in Brazil lower than 20 mg kg$^{-1}$. Similar results were obtained by Yesudhason et al. [57] that found an HIM ranging from 1 to 22.9 mg kg$^{-1}$ in 78.9% of 290 samples of canned products from Oman with an overall mean value of 3.18 mg kg$^{-1}$. HIM levels in the range 2.6–30.4 mg kg$^{-1}$ were reported by Er et al. [58] in Turkish canned tuna fish samples. Even worse, HIM content exceeding the limits of 50 mg kg$^{-1}$ set by the U.S. Food and Drug Administration, was found in 18.33% of canned fish samples marketed in Iran as reported by Peivasteh-Roudsari et al. [59], which also underlined the significant difference in HIM concentrations between canned tuna in oil and in brine. Among canned tuna, the only non-compliant sample with a concentration of 2219 mg kg$^{-1}$ was a "residual of the meal" associated with a scombroid poisoning case. This was a piece of tuna from a retail market that the seller had sold from a large open can and stored refrigerated for several days. Poisoning cases resulting from this common selling practice, already reported by Piersanti et al. [37], may be a risk for consumers health. As shown in Table 1, the highest HIM level found among canned mackerel batches was 105 ± 13 mg kg$^{-1}$. This value, associated with a compliant sample, taking into account measurement uncertainty, was higher than the maximum values reported for canned mackerel (24 mg kg$^{-1}$ and 19.1 mg kg$^{-1}$ reported by Tsai et al. and Petrovic et al., respectively [39,60]).

### 4.1.3. Ripened Products

As shown in Figure 2, all ripened products, in large part processed in Albania (43% of the total), were below the regulatory limits of concentrations lower than 50 mg kg$^{-1}$ in 90% of the samples. These findings are in contrast with the trend reported by Vosikis et al. [61] and in our previous survey [32]. In the past, the presence of HIM above limits in ripened anchovies was frequently the subject of several RASSF alert notifications and a matter for Commission Decision (EC) 642/2007 [62] regarding risk related to the consumption of fishery products imported from Albania. The low contamination level found in our survey among ripened products could be related to more efficient control activity plans undertaken by the EU for this kind of product.

### 4.2. Risk Exposure

As shown in Table 2, the highest intakes were registered for adolescent males, with percentages of NOAEL up to 12.1%, obtained for consumers of fresh and frozen fish. The lowest mean intake was obtained for children (female), for which the NOAEL percentage does not exceed 8.4% related to fresh and frozen fish consumers. Regarding the overall consumption of seafood, the global intake of HIM estimated for the global population follows the order: adolescents > adults > children > elderly, with NOAEL percentages ranging from 5.7–4.5%. Due to significantly lower consumption, consumers of preserved seafood have registered NOAEL percentages up to 35-fold lower than those of overall fish. In Table 3 the exposure assessment from fish consumption, under a most likely exposure scenario, is reported. Gram for gram (the most likely levels of HIM found in the survey were about 11, 1.6 and

15-fold lower than the average for overall, preserved and fresh/frozen seafood, respectively), the results confirmed the intake distribution obtained under an average exposure scenario. In particular, all NOAEL percentage intakes resulted in levels lower than 1%, with the highest value equal to 0.80% measured for male consumers of overall seafood. The highest mean intakes were registered for adolescent male consumers, with mean percentages of NOAEL corresponding to 0.78%, 0.19%, and 0.80% related to fresh/frozen, preserved, and overall seafood, respectively. According to the state verified under average exposure scenario, regarding the overall consumption of seafood, the global intake of HIM estimated for the global population follows the order: adolescents > adults > children > elderly, with NOAEL percentages ranging from 0.53–0.42%. Analogously, consumers of preserved seafood have registered NOAEL percentages significantly lower than those of overall seafood, in the order of about four-fold. This mean that the HIM intake difference between preserved and overall seafood consumers is less pronounced under the most likely exposure scenario, and that no substantial risk subsists relating to HIM intake from fish consumption. Regarding the specific exposure assessment developed for canned tuna consumption, the intake under average exposure scenario resulted equal to 0.03 mg Die$^{-1}$, corresponding to 0.07% of NOAEL. The values slightly increase under the most likely scenario, becoming equal to 0.04 mg Die$^{-1}$, corresponding to 0.08% of NOAEL. The results of the present exposure assessment are in agreement with those recently reported by Rahmani et al. in Iran [63] for canned tuna consumption, by Yesudhason et al. in Oman [57] for fresh and processed fish, and by Hariri et al. in Morocco [64] for chilled, frozen, canned, and semi-preserved fish. All these studies confirmed that, although control is needed due to possible acute poisoning phenomena, in most cases the post-catching and commercialization practices of fish are adequate, guaranteeing the good overall quality of fish.

## 5. Conclusions

A highly practical approach involving screening and confirmatory methods was used to efficiently determinate the histamine content in 474 samples of fish and fish products collected in Puglia and Basilicata regions (south part of Italy) during the years 2015–2019, in the framework of the official control. Although the results showed an increasing good quality of fish and a low risk exposure for the consumers, some scombroid poisoning cases, related to non-compliant tuna samples, confirmed the need for constant monitoring to avoid health risk. Furthermore, future studies will be oriented to investigate the presence of unauthorized additives to HIM-contaminated tuna samples, as required by the EU Commission.

**Author Contributions:** Conceptualization, S.L.M., G.M. and M.I.; methodology, S.L.M., G.M. and M.I.; software, S.L.M. and M.I.; validation, S.L.M., M.I. and S.S.; formal analysis, S.L.M., M.I. and S.S.; investigation, S.L.M., S.S. and P.D.; resources, S.S. and P.D.; data curation, S.L.M. and M.I.; writing—original draft preparation, S.L.M., M.I., S.S.; writing—review and editing, S.L.M. and M.I.; visualization, S.L.M. and M.I.; supervision, M.M. and A.E.C.; project administration, M.M.; funding acquisition, M.M. All authors have read and agreed to the published version of the manuscript.

**Funding:** This research received no external funding. Research and preparation of the article have been carried out entirely by internal resources of the "Istituto Zooprofilattico Sperimentale della Puglia e della Basilicata".

**Acknowledgments:** Istituto Zooprofilattico Sperimentale della Puglia e della Basilicata (Foggia, Italy) is thanked for providing financial support.

**Conflicts of Interest:** The authors declare no conflict of interest.

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
