# Peer review of "A 5-Years (2015–2019) Control Activity of an EU Laboratory: Contamination of Histamine in Fish Products and Exposure Assessment"

_applsci, doi:10.3390/app10238693_

Round 1

Reviewer 1 Report

Following issues should be solved in the manuscript:

  1. Lines 12-13: not so understandable sentence, please adjust it.
  2. lines 43-76: can be filled with certain data refereeing to the occurrence of histamine in different seafood species. The exact formation of histamine and microorganisms capable to decarboxylate histidine should be added too.
  3. The aim at the end of the Introduction part should be explained in more exact way.
  4. Section 2.2: HPLC/FLD method has to be described more in detail.
  5. Section 2.3: the section should be the part of the section 2.1. There is no reason for RISK EXPOSURE section in material and methods part.
  6. Section 2.2: the used statistical analyses were not almost described at all.
  7. Figures 1, 2 and 3: why statistical analysis was not performed, ANOVA, PCA etc.
  8. Tabl1 1: the statistical analysis was not performed.
  9. Tables 2 and 3: the source should be emphasized in titles of Tables.

Author Response

The authors would like to thank the reviewers for their effort in improving the scientific impact of the Paper. The manuscript has been revised, according to reviewers suggestions, editing corrections and rewording the text where necessary.

Reply to reviewer #1:  

Following issues should be solved in the manuscript:

- Lines 12-13: not so understandable sentence, please adjust it.

Answer: “Featured Application” paragraph was modified, substantially. According to the referee’s comment, the sentence has been simplified, in order to improve the readability (lines 16-18): “The exposure assessment was also developed. This elaboration gave useful parameters, also for other scientists who wish to carry out more extensive risk assessment studies”.

- Lines 43-76: can be filled with certain data refereeing to the occurrence of histamine in different seafood species. The exact formation of histamine and microorganisms capable to decarboxylate histidine should be added too.

Answer: As suggested, the required information about seafood species, histamine formation and microorganisms able to decarboxylate histidine was added (lines 46-58).

- The aim at the end of the Introduction part should be explained in more exact way.

Answer: The last part of Introduction, concerning the aim of this study was substantially re-elaborated in order to clarify the meaning (lines 102-108): “In this study, a contribution to the evaluation of exposure to HIM from fish consumption is reported. The results obtained by analyzing fresh, canned and ripened fish products for the determination of HIM are described and elaborated as risk exposure. The analyses were carried out within the official control plans in charge to the Istituto Zooprofilattico Sperimentale della Puglia e della Basilicata (IZS-PB) in the last five years (2015-2019)”.

- Section 2.2: HPLC/FLD method has to be described more in detail.

Answer: As requested by the referee, information about sample pre-treatment and a fully description of both ELISA and HPLC/FLD methods were added in Materials and Methods section (paragraphs from 2.2 to 2.5, lines 191-295).

- Section 2.3: the section should be the part of the section 2.1. There is no reason for RISK EXPOSURE section in material and methods part.

Answer: According to the referee’s suggestion, the section 2.3 was removed. The description of the approach used for elaborating the risk exposure was moved at the end of par. 2.1 (lines 130-159).

Section 2.2: the used statistical analyses were not almost described at all.

Answer: Statistical analysis was improved. One-way ANOVA and Student’s t-test were used to compare contamination levels of different types of seafood investigated. The results have been commented (lines 334-339) and reported as integration of table 1. Moreover, another comment and a new reference, related to the approach used for elaborating data lower than detection limit, have been added (lines 265-268).

Figures 1, 2 and 3: why statistical analysis was not performed, ANOVA, PCA etc.

Answer: As requested by the referee n.2, Figure 1 was removed. Figures 2 and 3 report a graphical elaboration of sample subdivision by concentration range. The number of samples which showed a concentration in the indicated range has been shown. Consequently, in this case no statistical analysis was performed. Figures 2 and 3 (now 1 and 2) have been completed by adding the labels on x-y-axis.

Table 1: the statistical analysis was not performed.

Answer: As per reviewer’s suggestion, the results obtained from statistical analysis have been added for integrating table 1.

Tables 2 and 3: the source should be emphasized in titles of Tables.

Answer: Following the reviewer’s suggestion, the source (INRAN-SCAI 2005-2006) has been added in the table headings.

Reviewer 2 Report

Somewhere is abbreviation for histamine (HIM) is used, somewhere not, please unify.

Matherials and Methods

It is not so clear 474 batches (3130 determinations) - how many samples from each batch and how many times were measured.

What about some statistical evaluation?

Results and Discussion

I miss statistical evaluation. 

There is no reference to Table 3 in the  text.

Conclusions 

meaningless, please adjust

Refernces

In some References are abreviated names of journal, in some References are full names of journal, please unify.

Author Response

The authors would like to thank the reviewer for his effort in improving the scientific impact of the Paper. The manuscript has been revised, according to reviewer's suggestions, editing corrections and rewording the text where necessary.

Reply to reviewer #2:

Somewhere is abbreviation for histamine (HIM) is used, somewhere not, please unify.

Answer: As suggested, abbreviation for histamine (HIM) was used in the whole text.

Materials and Method -It is not so clear 474 batches (3130 determinations) - how many samples from each batch and how many times were measured.

Answer:  In our survey 474 batches were analyzed. As specified in Materials and Method section, each batch was collected by the technicians of the local Health Service and border control authorities in nine sample units, as indicated in the Commission Regulation (EC) 2073/2005. However, in specific cases (suspected poisoning episode or sampling in the retail market) only a sample unit was taken. As reported at line 323 (in the paragraph 2.6  Data handling and statistical analysis), 332 batches were collected in nine sample units (and each sample unit was analyzed independently) and 142 batches were collected in single sample unit. The overall determinations were 3130 (142 + 332*9 = 3130).

For ELISA test each sample unit, standard solutions and quality control were pipetted twice into the microplate wells. This information is given in the lines 188-189.

For HPLC/FLD method each sample unit was analyzed in duplicate and results were calculated as the average of respective replicates. This information is given in the lines 245-246.

Materials and Method -What about some statistical evaluation?

Answer: Statistical analysis was improved. One-way ANOVA and Student’s t-test were used to compare contamination levels of different types of seafood investigated. See paragraph -2.6 Data handling and statistical analysis (lines 268-271). Moreover, another reference related to the approach used for elaborating data lower than detection limit has been added (lines 265-268).

Results and Discussion- I miss statistical evaluation.

Answer: Statistical analysis was completed, elaborating data by suing one-way ANOVA and Student’s t-test. The distribution of data obtained for different types of seafood were compared. The results have been commented (lines 334-339) and reported as integration of table 1.

Results and Discussion- There is no reference to Table 3 in the text.

Answer: At line 237 (previous version of article) Table 3 was indicated as Table 2, wrongly.

As suggested, the reference to Table 3 was added in the paragraph 3.2 - Risk exposure (line 485).

Conclusions- meaningless, please adjust

Answer: According to the referee’s comment, the paragraph Conclusion was modified. Some sentences have been simplified, in order to improve the readability.

References- In some References are abbreviated names of journal, in some References are full names of journal, please unify.

Answer: References list was carefully checked and drafted according to the authors guidelines. Unfortunately, the Journal’s name abbreviation for the Italian Journal of Food Safety is not available (lines 656-657).

Reviewer 3 Report

The study presents histamine contamination of fish products collected in Puglia and Basilicata regions.The measurements were done with the use of ELISA test and HPLC/FLD methods. In my opinion, presenting only the results of histamine concentration in the studied samples is not sufficient for the study to be published in Applied Sciences. 

Comments of the Reviewer:

  1. The Featured Application is not clear and not sufficient.
  2. There is no information of sample pre-treatment and about procedure of ELISA and HPLC/FLD analysis in Materials and Methods. The statement of the Authors: "Details about sample pre-treatment step and instrumental methods were reported elsewhere" is not sufficient and should not be given is articles.
  3. The methodology of statistical analysis given in Methods is described insufficiently. In such studies statistics is of high importance. The results of statistical analysis are not presented neither in figures and tables nor in text of the article.
  4. Figures and tables are of pure quality and are not prepared according to requirements. Figure 1 is not cited in the article and is not needed. The axis in figures are not described correctly.
  5. There is no discussion with other Authors (Discussion should be presented as a separate chapter according to the requirements of Applied Sciences). The Authors present only "raw" results without any comparison with other Researchers.
  6. In chapter Results and discussion - point 3.1.1. - line 174-187 - it is not discussion, this information should be given in introduction.

  In my opinion article titled:"A 5-years (2015-2019) control activity of an EU laboratory: contamination of histamine in fish products and exposure assessment" in the presented version should not be published in Applied Sciences.

Author Response

The authors would like to thank the reviewer for his effort in improving the scientific impact of the Paper. The manuscript has been revised, according to reviewer's suggestions, editing corrections and rewording the text where necessary.

Reply to reviewer 3

The study presents histamine contamination of fish products collected in Puglia and Basilicata regions. The measurements were done with the use of ELISA test and HPLC/FLD methods. In my opinion, presenting only the results of histamine concentration in the studied samples is not sufficient for the study to be published in Applied Sciences.

Comments of the Reviewer:

The Featured Application is not clear and not sufficient.

Answer: “Featured Application” paragraph was modified, substantially. According to the referee’s comment, the paragraph has been improved, adding comments about the significance and importance of histamine determination in food safety. The readability was also improved, simplifying the sentences (lines 10-19).

There is no information of sample pre-treatment and about procedure of ELISA and HPLC/FLD analysis in Materials and Methods. The statement of the Authors: "Details about sample pre-treatment step and instrumental methods were reported elsewhere" is not sufficient and should not be given is articles.

Answer:  As required, information about sample pre-treatment and a fully description of both ELISA and HPLC/FLD methods were added in Materials and Methods section (paragraphs from 2.2 to 2.5, lines 161-257).

The methodology of statistical analysis given in Methods is described insufficiently. In such studies statistics is of high importance. The results of statistical analysis are not presented neither in figures and tables nor in text of the article.

Answer: Statistical analysis was improved. One-way ANOVA and Student’s t-test were used to compare contamination levels of different types of seafood investigated (lines 268-271). The results have been commented (lines 334-339) and reported as integration of table 1. Moreover, another comment and a new reference, related to the approach used for elaborating data lower than detection limit, have been added (lines 265-268).

Figures and tables are of pure quality and are not prepared according to requirements. Figure 1 is not cited in the article and is not needed. The axis in figures are not described correctly.

Answer: As per reviewer’s comment, the figures and tables have been modified. Figure 1 has been removed. Figures 2 and 3 have been completed, specifying the titles in y and x-axis. The resolution of figures has also been checked, setting the resolution to 300 dpi. Regarding tables, the source used for data elaboration (seafood consumption) has been added as a footnote.

There is no discussion with other Authors (Discussion should be presented as a separate chapter according to the requirements of Applied Sciences). The Authors present only "raw" results without any comparison with other Researchers.

Answer: As suggested and according to the requirements of the Applied Science Journal, the Discussion was presented as a separate paragraph (paragraph 4) and the results were compared to those reported by other researchers, for each category of fish product.

In chapter Results and discussion - point 3.1.1. - line 174-187 - it is not discussion, this information should be given in introduction.

Answer: According to the referee suggestion, the information given at lines 174-187 (Results and discussion - point 3.1.1.) of the previous version of the article, was moved to Introduction (lines 83-92).

Round 2

Reviewer 1 Report

I would suggest only to include the following publication to the manuscript since it is describing and emphasizing the histamine issue concerning not traditionally consumed fish species in the European Union:

Đorđević, Đ., Buchtova, H., & Borkovcova, I. (2016). Estimation of amino acids profile and escolar fish consumption risks due to biogenic amines content fluctuations in vacuum skin packaging/VSP during cold storage. LWT-Food Science and Technology66, 657-663.

Author Response

Reply to reviewer #1:  I would suggest only to include the following publication to the manuscript since it is describing and emphasizing the histamine issue concerning not traditionally consumed fish species in the European Union:

Đorđević, Đ., Buchtova, H., & Borkovcova, I. (2016). Estimation of amino acids profile and escolar fish consumption risks due to biogenic amines content fluctuations in vacuum skin packaging/VSP during cold storage. LWT-Food Science and Technology, 66, 657-663.

Answer: The reference has been added in paragraph “4.1.1 Fresh, frozen ad defrosted fish” (lines 313-314) (reference No. 41).

The MS was spell-checked again and some keying errors have been corrected.

Reviewer 3 Report

Dear Authors,

As I have suggested in my previour review, presenting only the results of histamine concentration in the studied samples is not sufficient for the study to be published in Applied Sciences. Thank You for taking into account some improvements, but in my opinion the article in its present form should not be published in Applied Sciences.

Author Response

Reply to reviewer #3:  Dear Authors,

As I have suggested in my previour review, presenting only the results of histamine concentration in the studied samples is not sufficient for the study to be published in Applied Sciences. Thank You for taking into account some improvements, but in my opinion the article in its present form should not be published in Applied Sciences.

Answer: The authors believe that the article may be useful for scientists involved in seafood controls, since it reports an accurate description of the analytical approach (screening/confirmatory), a large amount of data and a comprehensive study of risk exposure.

For this reasons, the article was submitted for the Special Issue entitled “Advanced Analysis Techniques of Food Contaminants and Risk Assessment”.

The MS was spell-check again and some keying errors have been corrected.
